# Late Permian wood-borings reveal an intricate network of ecological relationships

Zhuo Feng [1,2,3], Jun Wang[2,4], Ronny Rößler [5,6], Adam Ślipiński[7] & Conrad Labandeira[8,9,10]

Beetles are the most diverse group of macroscopic organisms since the mid-Mesozoic. Much of beetle speciosity is attributable to myriad life habits, particularly diverse-feeding strategies involving interactions with plant substrates, such as wood. However, the life habits and early evolution of wood-boring beetles remain shrouded in mystery from a limited fossil record. Here we report new material from the upper Permian (Changhsingian Stage, ca. 254–252 million-years ago) of China documenting a microcosm of ecological associations involving a polyphagan wood-borer consuming cambial and wood tissues of the conifer *Ningxiaites specialis*. This earliest evidence for a component community of several trophically interacting taxa is frozen in time by exceptional preservation. The combination of an entry tunnel through bark, a cambium mother gallery, and up to 11 eggs placed in lateral niches—from which emerge multi-instar larval tunnels that consume cambium, wood and bark—is ecologically convergent with Early Cretaceous bark-beetle borings 120 million-years later.

[1] Institute of Deep Time Terrestrial Ecology, Yunnan University, Kunming 650091, China. [2] State Key Laboratory of Palaeobiology and Stratigraphy, Nanjing Institute of Geology and Palaeontology, Chinese Academy of Sciences, Nanjing 210008, China. [3] Institute of Karst Geology, Chinese Academy of Geological Sciences, Guilin 541004, China. [4] University of Chinese Academy of Sciences, Beijing 100049, China. [5] Museum für Naturkunde, Moritzstraße 20, D-09111 Chemnitz, Germany. [6] Geological Institute, TU Bergakademie Freiberg, Bernhard-von Cotta-Strasse 2, D-09599 Freiberg, Germany. [7] Australian National Insect Collection, CSIRO, GPO Box 1700, Canberra, ACT 2601, Australia. [8] Department of Paleobiology, Smithsonian Institution, Washington, DC 20013, USA. [9] Department of Entomology and BEES Program, University of Maryland, College Park, MD 20742, USA. [10] College of Life Sciences, Capital Normal University, Beijing 100048, China. Correspondence and requests for materials should be addressed to Z.F. (email: zhuofeng@ynu.edu.cn) or to C.L. (email: labandec@si.edu)

Wood-boring is an iconic feeding behaviour among extant terrestrial arthropods, particularly cambium and wood feeding insects and detritivorous oribatid mites[1–3]. Arthropod wood borers feed in roots, twigs, stems and trunks of dead or live woody plants, where they consume bark, phloem, sapwood and heartwood[1]. Although wood-boring has a long geological history traceable to the Late Devonian (ca. 383–359 million years ago, Ma), these occurrences are closely linked to the fossil wood record. Nearly all previous Palaeozoic records of wood-boring have been attributed principally to oribatid mites[3, 4]. The miniscule tunnels of oribatid mites likely represent the oldest form of wood boring, and provide indirect evidence for stereotypical tunnels and their contained coprolite clusters that display multimodal, instar-related, and size distributions[5, 6]. However, some larger borings have been found in pith tissues of Pennsylvanian-age tree-fern axes[6], which exhibit tunnel diameters approaching a tenfold increase compared to oribatid mites[3], consistent with a beetle fabricator[6] (Supplementary Note 1).

The sparse fossil record of beetle wood-borings begins with rare, poorly preserved, middle Permian (ca. 272–260 Ma) borings in permineralised, conifer-like wood showing simple tunnel structure oriented parallel to wood grain[7]. These borings occur in punky, fungally decayed wood[7, 8], and lack evidence for complex geometries that would indicate discrete, life-stage activities, such as broad galleries, laterally placed egg niches and subparallel larval tunnels within the cambial tissue layer[9, 10]. By the Late Triassic (ca. 237–201 Ma), cambium engravings are associated with the grazing of the secondary xylem (wood) outer surface and periderm (bark) inner surface[11–13]. These borings of wider diameter give rise to irregular, smaller, side-branch tunnels[12, 13]. Modern lineages of beetle wood borers, such as Buprestidae (jewel beetles), Cerambycidae (longhorn beetles), and various subgroups of Curculionidae (weevils), such as Scolytinae (bark beetles), currently are thought to have originated later, mostly during the Early Cretaceous (ca. 145–100 Ma)[14–16] (Supplementary Note 2).

In this study, we present detailed evidence for a wood-boring beetle that consumed both cambial and wood tissue of the conifer *Ningxiaites specialis*[17–19] and formed a complex tunnel geometry from the upper Permian of China. This discovery sheds light on the early evolutionary history of polyphagan beetles and their conifer hosts, and allows a better understanding of the ecological

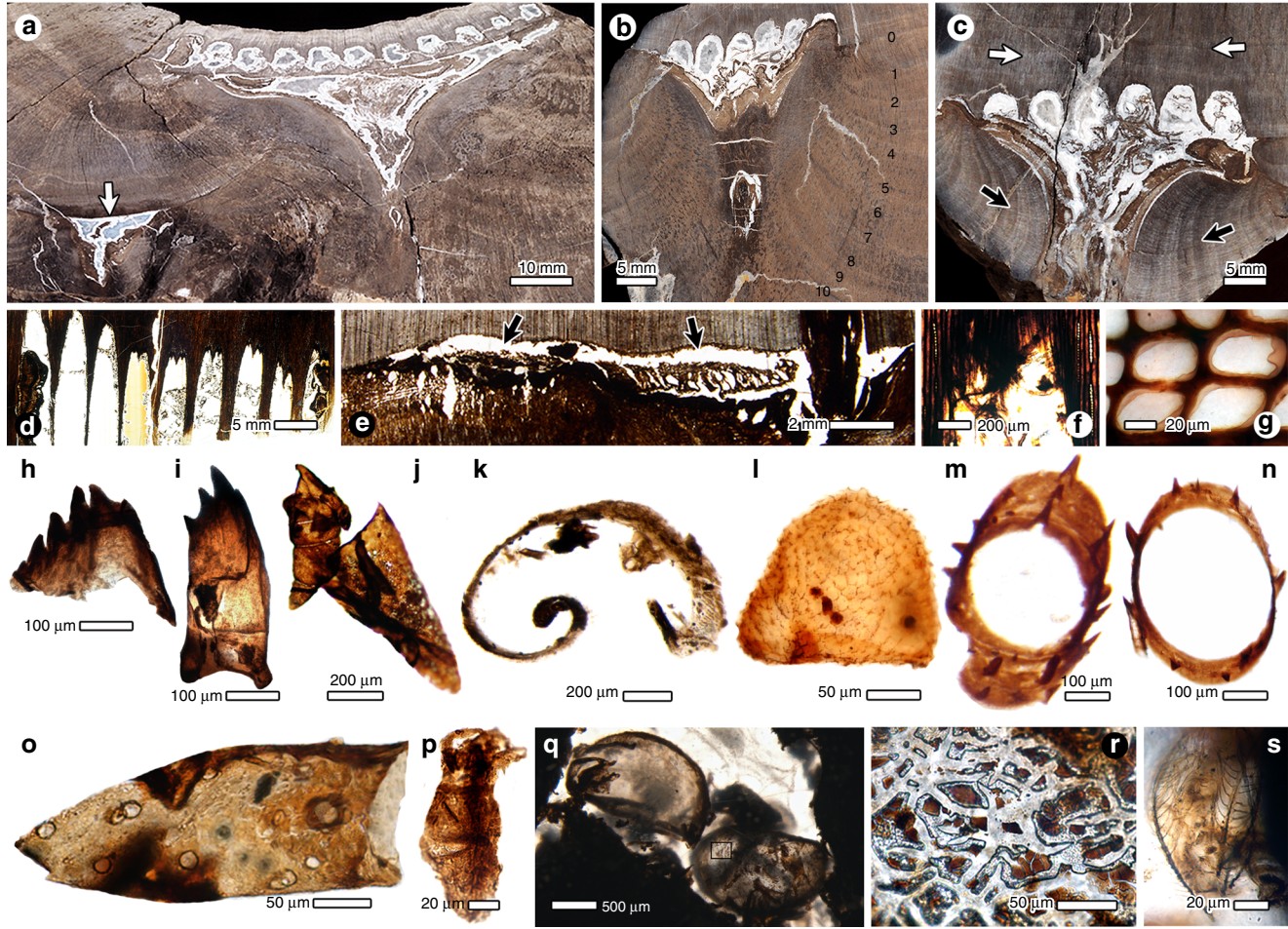

**Fig. 1** Beetle borings and their biological inclusions in conifer wood from the upper Permian of China. **a** Slightly oblique view of specimen YKLP20010, showing two networks of borings. At top is a series of 11 subsidiary (larval) tunnels and an associated triangular callus; the *arrow* indicates younger callus with borings embedded in wood. **b** YKLP20008a, with numbers indicating lateral growth rings postdating the borings. **c** YKLP20008b shows seven larval tunnels; *white arrows* indicate normal growth rings of the host tree prior to establishment of the borings; *black arrows* indicate growth rings were redirected to enclose the wound region. **d** YKLP20009, a longitudinal thin section through nine larval tunnels. **e** Section of YKLP2008a; *arrows* indicate the mother gallery between wood and bark. **f** Close-up of a larval tunnel, showing roughened tunnel wall texture attributable to fungal decay. **g** Tracheids displaying fungal induced cell-wall delamination. **h** Adult mandible. **i** Larval mandible. **j** Early instar mandible of a larval beetle. **k** A sclerite with a probable apodeme. **l** Egg chorion layer with mite microcoprolites on surface. **m**, **n** Spinose leg tibia or femur. **o** A chelicerate sclerite. **p** A pygmephorid mite. **q** Carapaces of large oribatid mites lodged in the larval tunnels. **r** Close-up of **q**, showing saprophagous fungi enveloping mite carapaces, marked by a square in **q**. **s** Fungal hyphae

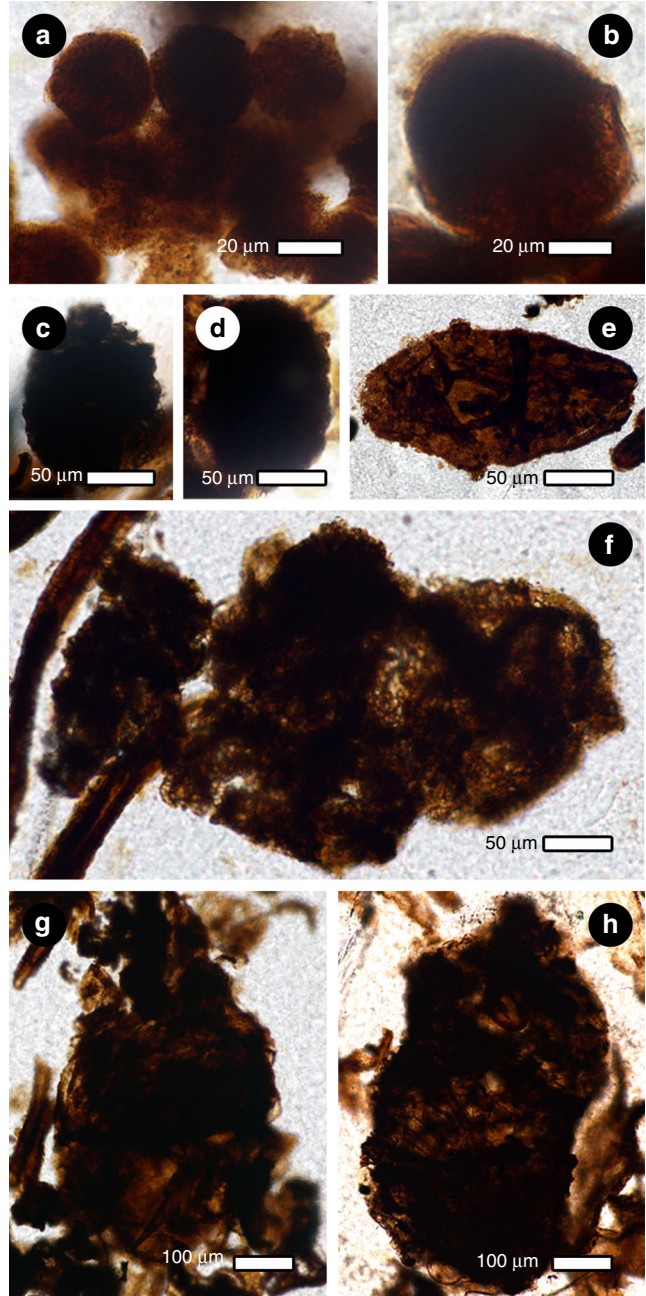

**Fig. 2** Four coprolite size classes produced by the late Permian wood-boring beetle from China. **a** Approximately eleven coprolites of size class I. **b** Single coprolite of size class II. **c–f** Coprolites of size class III, with cluster of four in **f**. **g**, **h** Coprolites of size class IV

relationships of multiple, trophically interacting taxa frozen in time by exceptional silica preservation.

## Results

**Morphology of wood borings versus beetle's life habit**. Several permineralised axes of the late Permian conifer wood *Ningxiaites specialis* host beetle borings (Fig. 1a–d and Supplementary Fig. 1a–e; Supplementary Note 3). These distinctive galleries and tunnels are divided into five sequential phases of beetle activity associated with a wood-boring life cycle. The first phase is an entry hole made through bark to access underlying cambial tissue. The hole and tunnel are rarely preserved, as later enrollment of subcortical callus (reaction wood) typically destroys the bark. The

second phase is the engraving of an extensive, linear mother gallery that is confluent with the cambium layer (Fig. 1e), and was the site where mating likely occurred. The third phase was oviposition of eggs by the female beetle singly into small, excavated niches at regular intervals along the topological upper surface of the horizontal mother gallery. The fourth phase was initiated by larvae hatched along a linear row of 6–11 niches, resulting in a series of parallel subsidiary (larval) tunnels that emerge at right angles from the mother gallery and progress upward along the cambium (Fig. 1d). Each tunnel underwent width expansions corresponding to larval moult growth increments deduced from coprolite size data (Fig. 2), and avoided breaches into adjacent tunnels. At mid-section along each larval tunnel, the trajectory descends into subjacent wood (Fig. 1d), leaving the cambium briefly, becoming completely embedded in wood, and eventually re-invading the cambium above to complete the larval tunnel before pupation. The fifth phase involves the exit tunnel and hole, inferred from biologies of modern wood-boring beetles[16, 20, 21], but evidently these were destroyed by callus enrollment that obliterated insect structures in the bark (Fig. 1a–e and Supplementary Fig. 1f, g). Wood tissue surrounding the borings displays remarkable fungal decay features (Fig. 1f, g).

**Arthropod remains in the borings**. Beetle remains (Fig. 1h–l and Supplementary Figs 1h and 2), beetle coprolites (Fig. 2), mycelia mats (Fig. 1r and Supplementary Fig. 4c–e), fungal hyphae (Fig. 1s), and other biotal elements commonly are preserved in the larval tunnels (Supplementary Note 3). The most prominent elements are isolated larval and adult mandibles representing multiple developmental stages (instars) of the wood-boring beetles (Fig. 1h, i and Supplementary Fig. 2a–f). These larval mandibles are broad, scoop shaped, with two basal condyles, four terminal incisors, a mesial oblique ridge and basal area lacking an identifiable mola (Fig. 1h). The adult mandible has three apical teeth and basal mola-like structure (Fig. 1i). Other arthropodan elements include parts of beetle larval mandibles possibly of another species (Fig. 1j), various body sclerites (Fig. 1k and Supplementary Fig. 2g–o), eggs displaying reticulate chorion microstructure (Fig. 1l and Supplementary Fig. 2p), falcate sclerites including a movable digit from a chelicera or pedipalp (Fig. 1o and Supplementary Fig. 3a–h), spinose segments of thoracic legs (Fig. 1m, n and Supplementary Fig. 3i–o), pygmephorid mites (Fig. 1p and Supplementary Fig. 4a), carapaces of oribatid mites (Fig. 1q and Supplementary Fig. 4b–e) and unidentifiable fragments (Supplementary Fig. 4f–k; Supplementary Note 3).

**Morphology of coprolites in the borings**. Coprolites occurring in larval tunnels are present as four size classes. These populations of coprolites were produced by beetle larval instars that contrast with much smaller mite coprolites (Fig. 1l). Beetle coprolites of size class I have the smallest diameters (< 50 μm), are subhexagonal in cross section, commonly clustered, and display finely comminuted but unidentifiable host-plant tissues (Fig. 2a). Size class II coprolites are somewhat larger in diameter and have contents similar to the first size class (Fig. 2b). Size class III coprolites are slightly < 150 μm in diameter, ovoidal to ellipsoidal, preserved singly (Fig. 2c–e) or clustered (Fig. 2f), and include tracheid fragments. Size class IV coprolites are up to 542 × 958 μm in dimension, generally preserved singly, typically ovoidal, and contain undigested tracheids (Fig. 2g, h). This size distribution indicates a larva with four instars[5, 6]. With the exception of identifiable tracheids, the opacity of the larger, beetle coprolites disallows identification of other constituents that would have been present in their diet, such as fungi.

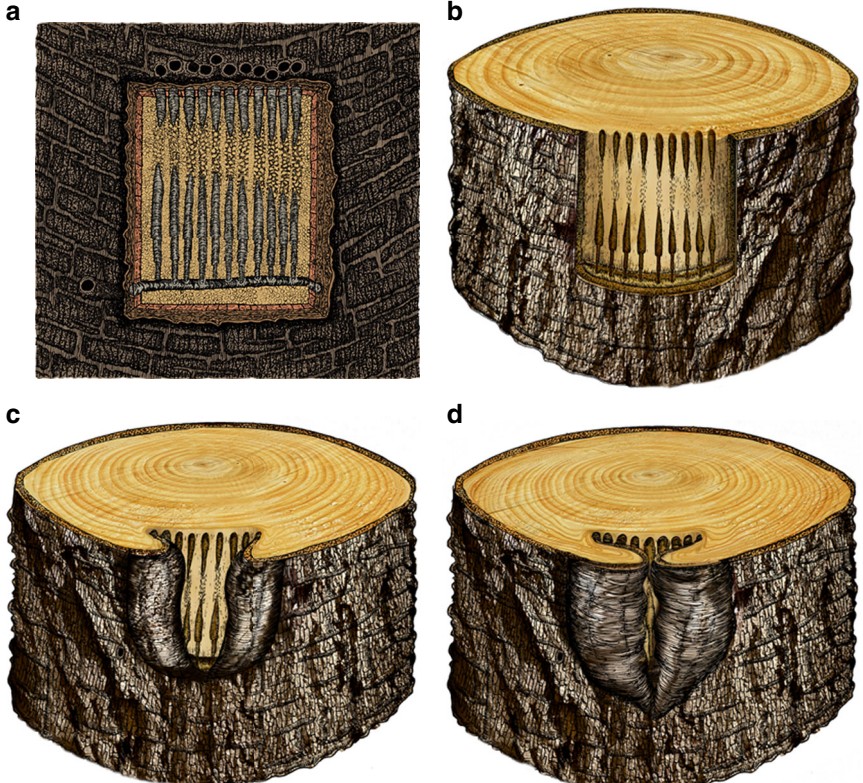

**Fig. 3** Reconstructions of a late Permian beetle boring in conifer wood from China. **a** Surface view of the beetle boring along the cambium in a conifer tree axis showing a horizontal mother gallery from which emerges vertical larval tunnels, each of which is characterised by larval width increases and invasion of the secondary xylem (wood) below; entry and exit holes inferred, based on destruction of tissue from wound closure by callus wood. **b**–**d** Three dimensional time-series reconstructions showing the formation of wood reaction tissue, eventual wound closure and resulting trunk scar. **b** Initial formation of the beetle boring consisting of a horizontal mother gallery and nine vertical larval tunnels. **c** At 5 years after the initial boring of **b**, there is considerable production of reaction wood resulting in partial wound closure; approximately three tunnels are observed on the wood surface, but six, unenclosed larval tunnels are seen along the transverse section at top. **d** At 10 years after the initial boring at **b**, the further production of additional reaction wood essentially has closed the wound, leaving a narrow strip of the wood outer surface exposed. Note significant development and deformation of annual ring structure at top, where, unenveloped larval tunnels are bordered by impinging, contorted wood. Reconstructions based on specimens YKLP2008a, YKLP2008b, YKLP2009 and YKLP20010 in Fig. 1

**Host plant wound reaction**. Other beetle borings observed in wood from the same tree trunk reveal that plant hosts were repeatedly attacked during their lifetimes (Fig. 1a–c; *arrow*). Malformations of callus wood and bark eventually enveloped the wound region that housed the beetle borings. The plant–host callus shows lateral growth increments that eventually covered the borings in a process lasting 10 or more years (Figs. 1b and 3). The resulting impingement wedge from enrolled callus commonly obliterated bark structures, such as entry and exit holes.

## Discussion

The beetle borer life cycle in *Ningxiaites* evidently underwent a dramatic dietary transformation. Adults initially bored through but did not consume indigestible bark, eventually engraving a mother gallery below to access nutritionally rich meristematic cambium. Larvae also consumed cambium during their early instar development, but fed on wood during a mid-instar interval. The entry of larvae into wood substrate required digestive processing either by cellulose-decomposing fungi in the tunnel or by a microbial biota in their gut. This was followed by rapid dietary shifts from wood to cambium to indigestible bark during the larva's last instar. Such dietary flexibility likely was a consequence of larvae domiciled in wood substrates early in beetle evolution.

The oldest presumed beetle fossils are from the Middle Pennsylvanian from northeastern Illinois, USA[22]. These insect fossils have

been variably interpreted and may be closest phylogenetically to extant Neuroptera (lacewings) based on wing venation[23]. Younger fossils of the Permian, however, provide sufficient morphological evidence that they can be attributed to early Coleoptera or closely related lineages[24]. The oldest Permian beetle fossils are the Tshekardocoleidae, Permocupedidae and related families of Archostemata, resembling modern Cupedidae (reticulated beetles). Larvae of Permian Archostemata likely inhabited rotten wood[8], and Permocupedidae have been implicated in construction of networks of coniferophyte borings from the Russian middle Permian[7]. Larval galleries from this specimen are poorly organised and interconnect with tunnels and shafts[7], similar to galleries of recent Cupedidae[8]. However, the disorganised, three-dimensional gallery architecture and absence of structurally distinct larval tunnels oriented perpendicular to the mother gallery does not fit the stereotypy of the *Ningxiaites* borer. Adult Permian and modern archostematans consist of a loosely intact, dorsoventrally flattened body with forwardly jutting mandibles[8, 25, 26], capable of making a flattened boring but not a tunnel circular in cross-section in wood[10, 20]. The behavioural and morphological evidence indicated by the late Permian tunnel and gallery system described herein indicates that one of the earliest lineages of the Polyphaga—one of the four major beetle clades—was present. This is a conclusion supported by late Permian body fossils affiliated with the Polyphaga on morphological grounds[24].

Wood-boring beetles evolved in association with microscopic fungi as food sources[20]. These fungi degraded woody debris lodged

in tunnels or were microbiotal elements in beetle's digestive tracts, thereby providing the important service of cellulose degradation[21]. Larval consumption of wood and the presence of cellulytic fungi strongly indicate obligate nutritional dependency on fungi and thus an early form of agriculture. The association of fungal hyphae within borings indicates that beetle occupants likely were primitive agriculturists, analogous to modern bark beetles, macrotermitine termites and attine ants[1, 27–30] that first appear during the Early Cretaceous. Diverse plant–fungus interactions are commonly known since the Devonian[31]. The oldest fungally infected wood has been reported in a Late Devonian progymnosperm[32], but lacks a wood-borer nutritional component. However, there is evidence for the third, arthropod component in a glossopterid–fungus–arthropod interaction nexus during the middle[33] and upper[34] Permian of Antarctica. The arthropod members of these tritrophic nutritional relationships probably were insects. The first persuasive evidence for a conifer–fungus–beetle wood-borer relationship was documented from the Middle Jurassic of Argentina[35] (Supplementary Note 1). The late Permian nutritional relationships described here occurred within a broader food web of a conifer source plant, fungal saprobes, insect and mite herbivores and fungivores, and chelicerate predators. We note that while fungi was present in the tunnels (Fig. 1s), neither spores nor hyphae were detected in the virtually opaque coprolites (Fig. 2), an observation also seen in the faecal pellets of extant oribatid mites and bark beetles that result from fluidisation of all but the most refractory ingested food prior to defecation[36, 37].

A related phenomenon is whether conditions were sufficient for the evolution of subsociality. Such conditions are: (i), presence of a shelter in the same environments where food is consumed; (ii), group living and cooperation; (iii), division of labour; and (iv), colony defense[38–41]. Our material appears to fulfil these requirements whereby networks of borings and their contents provide evidence for communal occupation of adult galleries, cultivation of fungi, division of labour between larvae and parents, and apparent colony defence from advertent obliteration of exit and entry holes through callus sealing (Fig. 3). Callus sealing is a structural defence mechanism occurring in modern conifers[1, 2], as it was in the late Permian *N. specialis*, although chemical defences for this fossil species, such as extensive resin production, has not been demonstrated. These features are consistent with subsociality. Because the anatomically distinctive fossil wood was collected in situ from the Sunjiagou Formation, which is accurately dated to the Changhsingian Stage of the late Permian (Supplementary Note 4), and direct evidence for insect sociality is absent from earlier deposits, our data currently indicate a late Permian date for the origin of subsociality in insects. Nevertheless, this interval of subsociality was fleeting and was extinguished by the global ecological crisis at the end of the Permian.

The evidence indicates the late Permian gallery and tunnel system lies considerably outside the range of the known behavioural, morphological and ecological features of Archostemata. We conclude these galleries are an extinct early lineage of Polyphaga, ecologically analogous to bark beetles whose earliest Early Cretaceous body-fossil occurrence postdates the late Permian by ca. 120 million-years. A Polyphaga assignment for the late Permian wood borer is supported by phylogenetic data, early Polyphaga presence in latest Permian deposits, and numerous Triassic earliest polyphagan body-fossil occurrences within 20 million-years after the late Permian[24, 42–46] (Supplementary Note 2). This distinctive life-habit of feeding relationships and stereotyped tunneling was probably a casualty of the end-Permian ecological crisis[45]. Evidence for the extinction of this beetle lineage includes the extinction of most beetle lineages arising from ecological devastation at the Permian–Triassic boundary[8, 24, 42, 46], reflected in the narrow diversity bottleneck of family-level beetle lineages at this event[47],

and also is consistent with phylogenetic studies[42], and a major Triassic–Jurassic time gap of simple and unelaborated wood borings[12, 13]. This distinctive life habit re-evolved later during the Early Cretaceous[48] by bark beetles[15, 42]. The demise of this unique wood-boring association among ancient conifers, fungi, beetles and other arthropods represent the beginnings of a food web centred in a closed, woody micro-environment that was soon extinguished but originated anew and expanded during the later Mesozoic[49]. This subsequent replacement attests to extensive evolutionary convergence seen with extant bark beetles[17, 21]. This transformation also parallels a similar Early Cretaceous turnover in a broad spectrum of plant–insect associations[50].

## Methods

**Locality and age**. All specimens of permineralised wood described herein were collected from the Sunjiagou Formation in Shitanjing Coalfield of Ningxia Huizu Autonomous Region, Northwest China. The bed from which the wood were retrieved is equivalent to the Changhsingian Stage, whose duration was ca. 252-254 million years ago (Supplementary Note 4).

**Preparation**. The wood specimens are preserved as three-dimensional silica permineralisations. The siliceous matrix in which the wood is embedded contains a large quantity of organic material that is responsible for excellent optical contrast under the microscope. Thin sections were made for detailed microscopical investigation by the following preparation methods. First, the specimens were cut into appropriately thin wafers by a diamond saw. The upper surface of each wafer was ground using a grinding wheel with fine-gained carborundum grit in a decreasing mesh series of #240, #400 and #800 grade sizes. The resulting smooth upper surface of each wafer was attached to a glass slide with Buehler EpoThin Epoxy Resin (20-8140-032) and EpoThin Epoxy Hardener (20-8142-016). The exposed surface subsequently was ground to a thickness of 30-50 μm.

**Imaging**. Photographs of hand specimens were imaged by a digital Nikon D3x camera with an AF-S Micro Nikkor 105 mm 1:2.8 G lens. Optical examination and photomicrographs were taken using a Leica DM 2500 M transmitted-light microscope and a Leica M 125 stereomicroscope, both equipped with a Leica DFC 500 digital camera. Composite images were stitched together with Adobe Photoshop CS5 Extended software.

**Repository**. Specimens, thin sections and digital photographs are housed in the Palaeobotanical Collections of the Institute of Deep Time Terrestrial Ecology, Yunnan University, China, with catalogue numbers YKLP20008, YKLP20009 and YKLP20010.

**Data availability**. No data sets were generated or analysed during the current study.

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

## Acknowledgements

This work was supported by the National Natural Science Foundation of China (41422201, 41672015, 41530101, 41372011), Strategic Priority Research Program (B) of the Chinese Academy of Sciences (XDB18000000, XDPB05), State Key Laboratory of Palaeobiology and Stratigraphy, Nanjing Institute of Geology and Palaeontology, CAS (20161101), Yunnan Provincial Science and Technology Department (2016FA019), China Geological Survey (DD20160061), Volkswagen Foundation (Az: I/84638), Deutsche Forschungsgemeinschaft (RO 1273/3-1) and Alexander von Humboldt-Foundation (1134259). We thank Bruce Halliday and Roy Norton for mite identifications, Michael Ivie for modern beetles, Jennifer Horne and Dongdong Zhang for Fig. 3. This is contribution 242 of the Evolution of Terrestrial Ecosystems Consortium at the National Museum of Natural History, Washington, DC.

## Author contributions

Z.F. designed the research, conducted field work and prepared fossil specimens. Z.F. and C.L. interpreted the data, prepared figures, and wrote the manuscript, with input from J.W., R.R. and A.Ś.

## Additional information

**Competing interests:** The authors declare no competing financial interests.

