## [Peer Review File · Nature Communications]

Reviewers' comments:

Reviewer #1 (Remarks to the Author):

Investigations of fossil animal-plant-fungal interactions have greatly expanded in the last 15 years revealing a wealth of previously unexplored data elucidating the history of herbivory and saprotrophic guilds and webs of complex multi-organism interactions that are finally allowing palaeontologists to reconstruct energy flows within ancient ecosystems with confidence. Moreover, much of this data, preserved as damage on plant fossils, provides some of the only fossil evidence to understand the stratigraphic ranges of key arthropod taxa for which body fossils may be scarce or unknown. That said, there are still many gaps in our knowledge of the groups of organisms involved in ancient interactions, the developmental series of specific damage types and the patterns of tissue consumption through time.

This study illustrates a remarkable example of damage to Late Permian conifer wood from China that is specifically attributable to wood-boring beetles. The results are unique for several reasons:

1. Most boring in Palaeozoic woods has been attributed previously to oribatid mites;
2. These are among the oldest confidently established beetle borings and certainly the oldest with complex tunnel geometry.
3. The damage in the Chinese woods shows successional development of the tunnel system apparently through several larval instars revealing partitioning of food resources and modifications to the feeding and digestive mechanisms through the animal's life cycle.
4. In contrast to almost all previous studies, this fossil example remarkably preserves fragments of the organisms involved in the damage to the wood (especially beetle and fungal parts), together with accessory biotic remains.
5. The fossil also reveals the host tree's response to damage via development of reaction tissues.

The paper is worthy of publication in the journal, but there are a few minor points that need cleaning up and/or clarifying before final publication. These include:

1. There are several minor typographic errors scattered through the text that need fixing. (see comments of the attached pdf). There are also some colloquial terms in the text and supplementary information (such as 'spotty' and 'scraggly') that might be better replaced/clarified for an international audience.
2. It would be good to point out early in the introduction that other arthropods apart from beetles were extensively involved in wood boring by the late Paleozoic; and later in the text, explain briefly or use references to note how one would consistently distinguish beetle vs mite borings. Some of this information is currently in the Supplementary Data, but I think it would be useful to present this as a brief summary in the main text.
3. The individual images in Fig. 2 are a little dark to see some of the details mentioned in the text (e.g., undigested tracheids). Could these images be lightened a little to show some more details of the coprolite contents without endowing them with excessive contrast?
4. The sentence on lines 173-175 may need some rephrasing. It could give the impression to readers that plant-fungal interactions are not known before the Late Permian. Further, there is some evidence of three-way fungal-plant-arthropod interactions from Middle and Late Permian woods from a couple of permineralized peat deposits in Antarctica (especially wood and root damage – Harper, Slater and other authors) – although perhaps these involved other invertebrate groups, such as mites; and there are other examples from the Triassic. The Chinese study would certainly extend the range of arthropod groups involved in such trophic interactions in the Permian, but it might be worth mentioning these previous studies if even to note their limitations.
5. Fig. 1R and S are quite small to decipher the details of the structures being illustrated. Perhaps this is a consequence of shrinkage during the pdf-making process, but could these images be enlarged to provide more detail?

Reviewer #3 (Remarks to the Author):

Dear authors,

this is an interesting and important manuscript on the early evolution of wood-boring beetles.

I have some additional comments:

Introduction: It would be important to start this section with a more general phrase on the earliest fossil record (body fossils as well as dates from phylogenetic reconstructions) of beetles, something like what you have written in line 168 to 171 in the supplement.

Results: The differentiation of five sequential phases of beetle activity is quite helpful for the reader. Unfortunately, these phases are not reflected in the figures or mentioned in the discussion in the main manuscript. Only in the supplemental text and Fig.S5 there is a connection to it. I would prefer to see a combined figure of Fig. 3 of the main text with figure S5 including an explanation of the main phases of beetle activity, as outlined in the beginning of the result paragraph.

Minor remarks:

Line 118 (main text): I guess you mean "pairs" instead of "pars" of beetle larval mandibles.

Line 141 (main text): "lasting ten or more years" - in the supplement (figure caption of S5) you only mention 8 years.

Line 181 (main text): I would prefer "environments" instead of "environs" - but I'm not a native speaker, of course.

Reviewer #4 (Remarks to the Author):

Wood borings have great potential to record in situ evidence for feeding behaviors of arthropods, which is among the most significant developments in the terrestrial ecosystem. Feng Zhuo and colleagues provide the earliest record of feeding trace, insect larvae and adults having been found in a fossil wood piece. They also make a nice and comprehensive summary of fossil wood borings and early polyphagan beetles. The study provides valuable insights into the early evolution of both insect-plant associations and beetles.

The paper is technically sound, with accurate descriptions. I don't have any objection to the details of the description. But the paper seems to be formatted in the Nature style. I suggest that the authors put the full descriptions of borings and their inclusions to the main text.

I have only two suggestions, which should be easily dealt with.

1 Modern wood-boring beetles (e.g. bark beetles) play a very important role in forest ecology, for example by creating complex early successional forest (e.g. Swanson et al., 2011, The forgotten stage of forest succession: early-successional ecosystems on forest sites. *Frontiers in Ecology and the Environment*). In my opinion, Permian wood-boring beetles probably have the similar role. I would suggest to provide a brief discussion about the potential significance of wood-boring beetles on the Permian forest ecology.

2 Line 199. "but was separated by an intervening Triassic and Jurassic record of simple and unelaborated borings". It is premature to say this considering so few records of Mesozoic wood borings. I suggest that the authors change it to "but was probably separated by an intervening Triassic and Jurassic record of simple and unelaborated borings".

In conclusion, the manuscript is of great interest and importance! I think that this paper merits publication in *Nature Communications*.

Bo Wang

Responses to Editor and Referees

We thank the referees for the constructive comments, and respond to them on a point-by-point basis (in blue text), below.

Reviewer #1 Remarks to the Author (Anonymous)

6. General comments. “Investigations of fossil animal-plant-fungal interactions have greatly expanded in the last 15 years revealing a wealth of previously unexplored data elucidating the history of herbivory and saprotrophic guilds and webs of complex multi-organism interactions that are finally allowing palaeontologists to reconstruct energy flows within ancient ecosystems with confidence. Moreover, much of this data, preserved as damage on plant fossils, provides some of the only fossil evidence to understand the stratigraphic ranges of key arthropod taxa for which body fossils may be scarce or unknown. That said, there are still many gaps in our knowledge of the groups of organisms involved in ancient interactions, the developmental series of specific damage types and the patterns of tissue consumption through time.

This study illustrates a remarkable example of damage to Late Permian conifer wood from China that is specifically attributable to wood-boring beetles. The results are unique for several reasons:

1. Most boring in Palaeozoic woods has been attributed previously to oribatid mites;
2. These are among the oldest confidently established beetle borings and certainly the oldest with complex tunnel geometry.
3. The damage in the Chinese woods shows successional development of the tunnel system apparently through several larval instars revealing partitioning of food resources and modifications to the feeding and digestive mechanisms through the animal's life cycle.
4. In contrast to almost all previous studies, this fossil example remarkably preserves fragments of the organisms involved in the damage to the wood (especially beetle and fungal parts), together with accessory biotic remains.
5. The fossil also reveals the host tree's response to damage via development of reaction tissues.”

AUTHOR'S RESPONSE: We agree and hope that this study presents an entirely new perspective to the establishment of a late Permian micro-community or organisms that inhabited a gallery-and-tunnel system in conifer wood.

7. Typographic errors and colloquial terms. “The paper is worthy of publication in the journal, but there are a few minor points that need cleaning up and/or clarifying before final publication. These include: [1.1], there are several minor typographic errors scattered through the text that need fixing. (See comments of the attached pdf). There are also some colloquial terms in the text and supplementary information (such as ‘spotty’ and ‘scraggly’) that might be better replaced/clarified for an international audience.”

AUTHOR'S RESPONSE: We have implemented the suggested changes by Referee #1 in the attached PDF file. In the Main Text the colloquial “spotty” has been changed to “sparse”. (See Main Text new line 89) In the Supplementary Information, “scraggly” has been changed to “... and a jagged appearance ...” (See Supplementary Information new lines 221–222)

8. Other late Paleozoic arthropod wood borers. “[1.2], it would be good to point out early in the introduction that other arthropods apart from beetles were extensively involved in wood boring by the late Paleozoic; and later in the text, explain briefly or use references to note how one would consistently distinguish beetle vs mite borings. Some of this information is currently in the Supplementary Data, but I think it would be useful to present this as a brief summary in the main text.”

AUTHOR’S RESPONSE: We have overhauled the introduction by inserting a new first paragraph that provides a broader context of Paleozoic wood borings and their arthropod fabricators. We broach the issue of distinguishing mite from beetle borings—a point that we discuss later in the manuscript. Our new introductory paragraph (Main Text new lines 76–88) is the following:

Wood-boring is an iconic feeding behaviour among extant terrestrial arthropods, particularly cambium and wood feeding insects and detritivorous oribatid mites^{1–3}. Arthropod wood borers feed in roots, twigs, stems and trunks of the dead or live woody plants, where they consume bark, phloem, sapwood and heartwood¹. Although wood-boring has a long geological history traceable to the Late Devonian (ca. 382–358 million years ago, Ma), these occurrences are closely linked to the fossil wood record. Nearly all previous Palaeozoic records of wood-boring have been attributed principally to oribatid mites^{3,4}. The miniscule tunnels of oribatid mites likely represent the oldest form of wood boring, and provide indirect evidence for stereotypical tunnels and their contained coprolite clusters that display multimodal, instar-related, size distributions^{5,6}. However, some larger borings have been found in pith tissues of Pennsylvanian-age tree-fern axis⁶, which exhibit tunnel diameters approaching a tenfold increase compared to oribatid mites³, consistent with a beetle fabricator⁴, and Supplementary Note 1.

The following new *Reference 5* has been linked to the new introductory paragraph above to better document the fossil record of Paleozoic wood-boring arthropods.

5. Falcon-Lang, H. J., Labandeira C. C. & Kirk, R. Herbivorous and detritivorous arthropod trace fossils associated with subhumid vegetation in the Middle Pennsylvanian of southern Britain. *Palaio* **30**, 192–206 (2015).

Also see Point 24 of Referee #3 below.

9. Lightening of photographic images. “[1.3], the individual images in Fig. 2 are a little dark to see some of the details mentioned in the text (e.g., undigested tracheids). Could these images be lightened a little to show some more details of the coprolite contents without endowing them with excessive contrast?”

AUTHOR’S RESPONSE: The images in the initial submission are of low-resolution according to the requirements of the Journal. In our re-submitted files images are of high-resolution, and the content of the coprolites are clearly visible.

10. Mentioning of previous studies indicating Permian plant–fungal relationships. “[1.4], the sentence on lines 173–175 may need some rephrasing. It could give the impression to readers

that plant-fungal interactions are not known before the Late Permian. Further, there is some evidence of three-way fungal-plant-arthropod interactions from Middle and Late Permian woods from a couple of permineralized peat deposits in Antarctica (especially wood and root damage – Harper, Slater and other authors) – although perhaps these involved other invertebrate groups, such as mites; and there are other examples from the Triassic. The Chinese study would certainly extend the range of arthropod groups involved in such trophic interactions in the Permian, but it might be worth mentioning these previous studies if even to note their limitations.”

AUTHOR’S RESPONSE: We have provided clarification to the subject of early plant–fungal–arthropod interactions by the following modification of text. (See Main Text new lines 207–215)

Diverse plant–fungus interactions are commonly known since the Devonian²⁹. The oldest fungally infected wood has been reported in a Late Devonian progymnosperm³⁰, but lacks a wood-borer nutritional component. However, there is evidence for the third, arthropod component in a glossopterid–fungus–arthropod interaction nexus during the middle³¹ and upper³² Permian of Antarctica. The arthropod members of these tritrophic nutritional relationships probably were insects. The first persuasive evidence for a conifer–fungus–beetle wood-borer relationship was documented from the Middle Jurassic of Argentina^{33, Supplementary Note 1}.

We have also added new references 31 and 32 linked to the above paragraph.

31. Slater, B. J., McLoughlin, S. & Hilton, J. Animal–plant interactions in a Middle Permian permineralised peat of the Bainmedart Coal Measures, Prince Charles Mountains, Antarctica. *Palaeogeogr., Palaeoclimatol., Palaeoecol.* **363–364**, 109–126 (2012).
32. Weaver, L., McLoughlin, S. & Drinnan, A. N. Fossil woods from the Upper Permian Bainmedart Coal Measures, northern Prince Charles Mountains, East Antarctica. *J. Austral. Geol. Geophys.* **16**, 655–676.

11. Enlargement of Figures 1R and 1S. “[1.5] Fig. 1R and S are quite small to decipher the details of the structures being illustrated. Perhaps this is a consequence of shrinkage during the pdf-making process, but could these images be enlarged to provide more detail?”

AUTHOR’S RESPONSE: The images in the initial submission are of low-resolution, based on the requirements of *Nature Communications*. In our current, re-submitted files the corresponding images are of higher-resolution, allowing detailed structures of the specimens to be clearly visible.

Reviewer #3 Remarks to the Author (Anonymous)

23. General comments. [3.1], “Dear authors, this is an interesting and important manuscript on the early evolution of wood-boring beetles. I have some additional comments.”

AUTHOR’S RESPONSE: We agree and hope that this study presents an entirely new perspective to the establishment of a late Permian micro-community or organisms that inhabited a gallery-and-tunnel system.

24. Introduction set-up. [3.2], “Introduction: It would be important to start this section with a more general phrase on the earliest fossil record (body fossils as well as dates from phylogenetic reconstructions) of beetles, something like what you have written in line 168 to 171 in the supplement.”

AUTHOR’S RESPONSE: We have overhauled the introduction by inserting a new first paragraph that provides a broader context of Paleozoic wood borings and their arthropod fabricators. We broach the issue of distinguishing mite from beetle borings—a point that we discuss later in the manuscript. Our new introductory paragraph (Main Text new lines 76–88) is the following:

Wood-boring is an iconic feeding behaviour among extant terrestrial arthropods, particularly cambium and wood feeding insects and detritivorous oribatid mites^{1–3}. Arthropod wood borers feed in roots, twigs, stems and trunks of the dead or live woody plants, where they consume bark, phloem, sapwood and heartwood¹. Although wood-boring has a long geological history traceable to the Late Devonian (ca. 382–358 million years ago, Ma), these occurrences are closely linked to the fossil wood record. Nearly all previous Palaeozoic records of wood-boring have been attributed principally to oribatid mites^{3,4}. The miniscule tunnels of oribatid mites likely represent the oldest form of wood boring, and provide indirect evidence for stereotypical tunnels and their contained coprolite clusters that display multimodal, instar-related, size distributions^{5,6}. However, some larger borings have been found in pith tissues of Pennsylvanian-age tree-fern axis⁶, which exhibit tunnel diameters approaching a tenfold increase compared to oribatid mites³, consistent with a beetle fabricator⁴, and Supplementary Note 1.

The following new *Reference 5* has been linked to the new introductory paragraph above to better document the fossil record of Paleozoic wood-boring arthropods.

5. Falcon-Lang, H. J., Labandeira C. C. & Kirk, R. Herbivorous and detritivorous arthropod trace fossils associated with subhumid vegetation in the Middle Pennsylvanian of southern Britain. *Palaios* **30**, 192–206 (2015).

Also see Point 8 of Reviewer #1 above.

25. Better explanation of the five sequential phases of beetle activity. [3.3], “Results: The differentiation of five sequential phases of beetle activity is quite helpful for the reader. Unfortunately, these phases are not reflected in the figures or mentioned in the discussion in the main manuscript. Only in the supplemental text and Fig.S5 there is a connection to it. I would

prefer to see a combined figure of Fig. 3 of the main text with figure S5 including an explanation of the main phases of beetle activity, as outlined in the beginning of the result paragraph.”

AUTHOR’S RESPONSE: We now have moved Fig. S5 from the Supplementary Information to the Main Text. The new Fig. 3 of the Main Text illustrates the reconstruction of the total beetle gallery and tunnel system (now Fig. 3a). In addition, the Main Text also refers to the three reconstructions of the beetle gallery and tunnel system at the initial formation of beetle activity (now Fig. 3b), at five years after the initial beetle activity (now Fig. 3c), and at ten years after initial beetle activity (now Fig. 3d). The illustrations of Fig. 3 also have been linked to the main text to better flesh out what is discussed. We also make reference to Supplementary Note 3 of the Supplementary Information (“Full Descriptions of the Borings and their Inclusions”) that provides more of the gory details of the borings in four specimens that are not detailed in the Main Text.

26. Some minor remarks. [3.4], “Minor remarks: Line 118 (main text): I guess you mean "pairs" instead of "pars" of beetle larval mandibles.”

AUTHOR’S RESPONSE: The word “pars” now has been changed to “parts”. See Main Text new line 140.

27. Eight or ten years? [3.5], “Line 141 (main text): "lasting ten or more years" - in the supplement (figure caption of S5) you only mention 8 years.”

AUTHOR’S RESPONSE: The time-series reconstructions of previous Fig. S5 have been combined into the new, revised Fig. 3. The eight years mentioned in the previous figure caption was a mistake and now has been changed to ten years, matching the text. (See Main Text new lines 166 and 475).

28. “Environments”, instead of “environs”? [3.6], Line 181 (main text): I would prefer "environments" instead of "environs" - but I’m not a native speaker, of course.

AUTHOR’S RESPONSE: The previous term “environs” has been changed to “micro-environments”, given that a limited community of small organisms inhabiting a wood-boring is being discussed. (See Main Text new line 257).

Reviewer #4 Remarks to the Author (Dr. Bo Wang)

29. General comments. “Wood borings have great potential to record in situ evidence for feeding behaviors of arthropods, which is among the most significant developments in the terrestrial ecosystem. Feng Zhuo and colleagues provide the earliest record of feeding trace, insect larvae and adults having been found in a fossil wood piece. They also make a nice and comprehensive summary of fossil wood borings and early polyphagan beetles. The study provides valuable insights into the early evolution of both insect-plant associations and beetles.”

AUTHOR’S RESPONSE: We agree and hope that this study presents an entirely new perspective to the establishment of a late Permian micro-community or organisms that inhabited a gallery-and-tunnel system.

30. Full descriptions of borings moved to the main text. “The paper is technically sound, with accurate descriptions. I don't have any objection to the details of the description. But the paper seems to be formatted in the Nature style. I suggest that the authors put the full descriptions of borings and their inclusions to the main text.”

AUTHOR'S RESPONSE: We have moved some of the material from the Supplementary Information into the Main Text, such as Fig. S5, which is now Fig. 3. The introductory paragraph that now forms the Main Text also has been moved from the Supplementary Information. However, we have left the descriptions of each of the four specimens of wood with beetle borings in the Supplementary Information, recognising that their major features involving beetle borer biology are mentioned in the Main Text.

31. First suggestion: linking beetles to late Permian forest ecology. “I have only two suggestions, which should be easily dealt with. [4.1], Modern wood-boring beetles (e.g. bark beetles) play a very important role in forest ecology, for example by creating complex early successional forest (e.g. Swanson et al., 2011, The forgotten stage of forest succession: early-successional ecosystems on forest sites. *Frontiers in Ecology and the Environment*). In my opinion, Permian wood-boring beetles probably have the similar role. I would suggest to provide a brief discussion about the potential significance of wood-boring beetles on the Permian forest ecology.”

AUTHOR'S RESPONSE: This is an important suggestion. Extant wood-boring beetles certainly play a very important role in shaping forest ecosystems. And they probably also played similar role in the Permian forest ecosystems. However, we do discuss the role of the potential significance of beetle's borings on the Permian forest ecology, its extirpation, and its reappearance during the mid-Mesozoic. This role is discussed in the following added sentence (Main Text new lines 255–258.)

The demise of this unique wood-boring association among ancient conifers, fungi, beetles and other arthropods represent the beginnings of a food web centered in a closed, woody micro-environment that was soon extinguished but originated a new and expanded during the later Mesozoic⁴⁷.

Reference 47 was added, at the suggestion of Dr. Wang, to source the above text.

47. Swanson, M. E. et al. The forgotten stage of forest succession: early-successional ecosystems on forest sites. *Front. Ecol. Environ.* **9**, doi:10.1890/090157 (2010).

32. Second suggestion: interpreting the record of Mesozoic wood borings. [4.2], Line 199. “but was separated by an intervening Triassic and Jurassic record of simple and unelaborated borings”. It is premature to say this considering so few records of Mesozoic wood borings. I suggest that the authors change it to “but was [probably] separated by an intervening Triassic and Jurassic record of simple and unelaborated borings”.

AUTHOR'S RESPONSE: The sentence has been rephrased as following, based on this important suggestion (See Main Text new lines 248–253):

Evidence for the extinction of this beetle lineage includes the extinction of most beetle lineages arising from the ecological devastation at the Permian–Triassic boundary^{8,23,40,44}, reflected in the narrow diversity bottleneck of family-level beetle lineages at the Permian–Triassic boundary⁴⁵, and also is consistent with phylogenetic studies⁴⁰, and a major Triassic–Jurassic time gap of simple and unelaborated wood borings^{11,13}.

Other Modifications

33. Acknowledgements

AUTHOR'S RESPONSE: We have added the following sentence (Main Text new lines 420–422) to the Acknowledgements section: “Bo Wang, three anonymous reviewers, and Editor Izzadora Andrew provided very helpful comments that significantly improved this contribution.”

34. Graphic Abstract

AUTHOR'S RESPONSE: We have prepared a Graphic Abstract as an attached image file and offer a caption (assuming one is needed) at the end of the Main Text (new lines 479–483):

Food web showing trophic pathways of a micro-community of organisms in a conifer wood from the upper Permian of China, some 254–252 million years ago. Arrow widths of consumer→consumed trophic paths approximately proportional to assimilated biomass flow, The 1 mm scale bar at lower left applies to the central panel of conifer wood and its tunnel-inhabiting constituents.

Many thanks for your consideration and Best wishes

REVIEWERS' COMMENTS:

Reviewer #1 (Remarks to the Author):

The authors have now emended their manuscript to a form suitable for publication in Nature Communications. The paper documents a novel assemblage of plant-animal-fungal interactions and makes a strong advance in our understanding of the development of subsociality and the partitioning of food resources through a life cycle in the insect fossil record. There are just a few minor cosmetic changes that I suggest making before final acceptance. Line numbers refer to those on the pdf.

Main text:

Line 257. Change 'a new' to 'anew'

Line 450. Perhaps 'delamination' would be a better word than 'alternation' here.

Line 455. Fig. 1.S. is stated to be fungal hyphae, but the filaments in the illustration appear to be way too regularly pinnate for fungal hyphae. Could they be arthropod setae/antennae?

Supplementary Notes:

Line 30. Change 'have' to 'has'.

Line 41. Change 'are early Permian age' to 'are early Permian in age'.

Line 55. Fossil woods previously referred to Araucarioxylon are now better assigned to Agathoxylon owing to issues of taxonomic priority. See Rößler, R., Philippe, M., van Konijnenburg-van Cittert, J.H.A., McLoughlin, S., Sakala, J., Zijlstra, G. & 35 others, 2014. Which name(s) should be used for Araucaria-like fossil wood? – Results of a poll. *Taxon* 63, 177–184.

Line 240. Change 'a cheliceral sclerites, attributable to chelicerate arthropod' to 'sclerites attributable to a chelicerate arthropod'.

Reviewer #3 (Remarks to the Author):

Dear authors,
thank you for your comments. I think the manuscript is suitable for publication now.

Reviewer #4 (Remarks to the Author):

The authors have successfully addressed all my concerns, improving the manuscript with their edits. This will be a fine contribution as it stands. I recommend acceptance in this form.

Responses to Referees

We thank the referees for the constructive comments, and respond to them on a point-by-point basis (in blue text), below.

Reviewer #1 (Remarks to the Author):

The authors have now emended their manuscript to a form suitable for publication in Nature Communications. The paper documents a novel assemblage of plant-animal-fungal interactions and makes a strong advance in our understanding of the development of subsociality and the partitioning of food resources through a life cycle in the insect fossil record. There are just a few minor cosmetic changes that I suggest making before final acceptance. Line numbers refer to those on the submitted pdf.

AUTHOR'S RESPONSE: We note that Referee #1 is satisfied with our revised manuscript. Few minor points raised by Referee #1 are listed below:

Main text:

Line 257. Change 'a new' to 'anew'

AUTHOR'S RESPONSE: "a new" was changed to "anew". (see new Line 214).

Line 450. Perhaps 'delamination' would be a better word than 'alternation' here.

AUTHOR'S RESPONSE: "alternation" was replaced by "delamination". (see new Line 403).

Line 455. Fig. 1.S. is stated to be fungal hyphae, but the filaments in the illustration appear to be way too regularly pinnate for fungal hyphae. Could they be arthropod setae/antennae?

AUTHOR'S RESPONSE: We believe the septate filaments in Figure 1s are fungal hyphae. We also have addressed the identity of the fungus in Supplementary Note 3 of the Supplementary Information in lines 254-260. Morphologically, the current filaments are nearly identical with the extant deuteromycetous species *Melanographium spinulosum* that described in the following papers:

Hughes, S. J. Revisiones Hyphomycetum aliquot cum appendice de nominibus rejiciendis. *Can. J. Bot.* **36**, 727–836 (1958).

Morris, E. F. The synnematosous genera of the Fungi Imperfecti. Series in the Biological Sciences, no. 3 (Western Illinois Uni. Press, 1963).

Supplementary Notes:

Line 30. Change 'have' to 'has'.

AUTHOR'S RESPONSE: “have” was changed to “has”. (see new Line 28).

Line 41. Change 'are early Permian age' to 'are early Permian in age'.

AUTHOR'S RESPONSE: “are early Permian age” was changed to “are early Permian in age”. (see new Line 39).

Line 55. Fossil woods previously referred to *Araucarioxylon* are now better assigned to *Agathoxylon* owing to issues of taxonomic priority. See Rößler, R., Philippe, M., van Konijnenburg-van Cittert, J.H.A., McLoughlin, S., Sakala, J., Zijlstra, G. & 35 others, 2014. Which name(s) should be used for *Araucaria*-like fossil wood? – Results of a poll. *Taxon* 63, 177–184.

AUTHOR'S RESPONSE: “*Araucarioxylon*” was changed to “*Agathoxylon*”. (see new Line 53).

Line 240. Change 'a cheliceral sclerites, attributable to chelicerate arthropod' to 'sclerites attributable to a chelicerate arthropod'.

AUTHOR'S RESPONSE: “a cheliceral sclerites, attributable to chelicerate arthropod” was changed to “sclerites attributable to a chelicerate arthropod”. (see new Line 238).

Reviewer #3 (Remarks to the Author):

Dear authors,

thank you for your comments. I think the manuscript is suitable for publication now.

AUTHOR'S RESPONSE: We note that Referee #3 appears satisfied with our revised manuscript.

Reviewer #4 (Remarks to the Author):

The authors have successfully addressed all my concerns, improving the manuscript with their edits. This will be a fine contribution as it stands. I recommend acceptance in this form.

AUTHOR'S RESPONSE: We also note that Referee #4 is also satisfied with our revised manuscript.

Many thanks for your consideration and Best wishes